# The Potential Impacts of Statins and Beta-Blockers on West Virginia Ichthyofauna

Joseph W. Kingsbury * and Kyle J. Hartman *

Davis College of Agriculture, Natural Resources and Design, West Virginia University, Morgantown, WV 26506-6125, USA

* Correspondence: joseph.kingsbury@mail.wvu.edu (J.W.K.); kyle.hartman@mail.wvu.edu (K.J.H.)

**Abstract:** Pharmaceuticals and personal care products (PPCPs), such as statins and beta-blockers, are commonly used to treat cardiovascular disease in adults. Active versions of these pharmaceuticals and their various metabolites enter surface waters via wastewater treatment plant (WWTP) discharge, as well as from other point sources. Sub-lethal effects of statins and beta-blockers on wild fish at environmental concentrations have been understudied up to this point. The objectives of this study were to use several health condition metrics and determine if there was a relationship between fish condition and environment concentrations of statins and beta-blockers near two West Virginia WWTPs. Water samples were collected from upstream, downstream, and effluent pipe from August to November 2022, and analyzed for atorvastatin, simvastatin, metoprolol, and carvedilol via liquid chromatography with tandem mass spectrometry. Fish were sampled upstream, at the discharge, and downstream of each WWTP in November 2022. Fish health was assessed with three metrics: relative weight (Wr), hepatosomatic index (HSI), and gonadosomatic index (GSI). ANOVAs were used to assess differences among the health metrics based on sex, genus/species, and location relative to WWTPs. Additionally, changes in Wr relative to surface water concentrations of statins and beta-blockers was modeled with a Bayesian linear mixed effects model, with surface water concentrations as fixed effects with a random slope, while the section and genus parameters were treated as random intercepts. Surface concentrations for atorvastatin (0.47–4.36 ng/L), simvastatin (0.27–0.95 ng/L), metoprolol (2.80–21.01 ng/L), and carvedilol (0.43–0.90 ng/L) varied across sampling sections. HSI based on sex and species were nearly significant. GSI was significantly higher in females. Wr differed among genera, as well as the interaction between genus and sample section ($p < 0.001$). Fixed effects from the linear mixed effects model showed Wr was negatively related to simvastatin ($-0.139$ [$-2.072$–$1.784$]) and carvedilol ($-0.262$ [$-2.164$–$1.682$]) while atorvastatin ($0.207$ [$-1.371$–$1.845$]) and metoprolol ($0.052$ [$-0.533$–$0.584$]) were positively related to Wr. Individual genera responded differently to each pharmaceutical based on location, indicating that it is likely that other factors were also influencing the fish health metrics. Further research targeting individual tissues and controlled experiments with different exposure regimes will be required to further enlighten the long-term effects of cardiovascular PPCPs on fish health.

**Keywords:** pharmaceuticals; fish health; ecotoxicology

## 1. Introduction

Pharmaceuticals, pharmaceutical metabolites, and other personal care products are found in surface and ground waters throughout the world, including the streams and rivers in the United States [1,2]. The ecotoxicological effects of these pharmaceutical pollutants on aquatic organisms are of concern due to the increasing usage of pharmaceuticals in human and veterinary medicine. Pharmaceuticals are a unique pollutant, because they are specifically designed to target specific physiological processes and persist for some time within the body. Another issue of concern specific to aquatic organisms is that they can be exposed to these pollutants for their entire lives, and multiple generations will

most likely be exposed as well. Acute toxicity exposure in the wild is highly unlikely and only applicable following a large accidental discharge. Thus, chronic toxicity studies are more relevant in assessing potential impacts to aquatic organisms [1,3]. The literature surrounding the acute toxicity of specific pharmaceuticals far outweighs the literature regarding their chronic or lifelong toxicity [4]. For the reasons outlined above, a brief review of known chronic toxicity effects regarding statins and beta-blockers on aquatic organisms is likely to be relative to scenarios found in most areas of interest.

Statins are lipid lowering agents that inhibit cholesterol synthesis by inhibiting the 3-hydroxymethlglutaril coenzyme A reductase (HMGR) enzyme. Statins are HMGR inhibitors in both vertebrates and in arthropods, but arthropods synthesize cholesterol via a slightly different mechanism [5,6]. A key concern regarding statins is their low log Kow values, indicating that they are likely to bioaccumulate, which can have wide ranging effects [7]. Chronic toxicity studies of statins are rare in general and almost non-existent regarding freshwater systems, with most current chronic toxicity studies having focused on marine organisms and arthropods. Based on the limited studies available, simvastatin has been shown to impact the reproduction and growth of crustaceans as well as several other invertebrates [6,8–10]. To the best of our knowledge, atorvastatin has few chronic toxicity studies with only one aquatic organism paper published by Santos et al. [7] hypothesizing that metazoans could be at risk to statins. There are currently no published studies regarding long-term or chronic exposure in fish species. Fish conditions relative to environmental concentrations within a controlled environment or within an observational study framework has not been published, to the best of our knowledge.

Beta-blockers are antihypertensive medications that inhibit beta-adrenergic receptors to control blood pressure, heart rate, and airway strength reactivity, depending on the location of the targeted beta-adrenergic receptors [11]. Just like mammals, fish also possess beta-adrenergic receptors in the heart, liver, and reproductive system, and are potentially susceptible to their effects [4,12,13]. Several studies have investigated the effects of beta-blockers on fish and found that fish are less susceptible than other aquatic organisms such as macroinvertebrates, plants, and algae [4]. However, a study by Triebskorn et al. [14] showed that chronic exposure to metoprolol at environmentally relevant levels can lead to ultrastructural changes in the gills, kidneys, and liver. Another study showed that prolonged exposure to beta-blockers can negatively impact fish growth, as well as decrease egg production [13]. Invertebrates do not possess beta-receptors like fish and mammals, but are still susceptible to their impacts via different mechanisms, such as membrane disruption [15]. Massarky et al. [16] hypothesized the potential for beta-blockers to be endocrine disruptors and that beta-blockers would affect the stress response of aquatic organisms, especially fish. The ecotoxicological impacts from carvedilol are not documented at all as of this time and need significant research. Chronic toxicity testing on macroinvertebrates and aquatic plants are still lacking, and multi-generational testing for beta-blockers is also quite scarce for all aquatic organisms. Congruently, bioaccumulation and field studies for the detection of beta-blockers in aquatic biota are also scarce.

There is a well-established understanding of the point sources for pharmaceutical waste [17] and the primary mechanism of action for both statins and beta-blockers within aquatic organisms are thought to be similar to mammals [18,19]. However, there is insufficient information regarding their long-term impacts or even the most effective approach for investigating these impacts. Previous studies have thoroughly investigated the acute toxicity of isolated pharmaceuticals in laboratory environments. In contrast, the number of chronic toxicity studies pales in comparison, and is an area of study that requires more attention going forward. In addition, mixture studies are sparse at best and provide the best possibility of simulating real-world exposure scenarios for aquatic organisms. We know that these pharmaceuticals pseudo-persist in the environment as a complex mixture, and initial mixture studies have shown that they can interact with one another leading to greater toxic effects on aquatic organisms [9,20,21]. Additional mixture studies and the various environmental factors that contribute to variations in pharmaceutical waste toxicity are still

required. At the same time, most impact studies involving aquatic organisms take place in laboratory environments, and there is a lack of field studies using real world systems to investigate the ecological impacts of pharmaceuticals (Table 1). Furthermore, information is lacking regarding the potential sublethal impacts pharmaceuticals have on growth, reproduction, and survival across multiple species and organisms over multiple generations. While direct impacts to a single species may not be harmful, impacts to multiple organisms or species throughout the food web at various trophic levels could potentially lead to ripple effects that are harder to predict and thus require further investigation. To gain a better understanding of the potential effects from these pharmaceuticals, it is essential that studies include multiple species simultaneously. By filling in these knowledge gaps, more accurate and effective risk assessments can be created, thereby giving managers better tools for assessing, reducing, and/or preventing severe ecological damage from pharmaceutical and personal care product wastes.

**Table 1.** Relevant research papers over the last 5 years regarding the toxicity and physiological impacts of various statins and beta-blockers on fish species.

| Pharmaceutical(s) | Therapeutic Class | Physiological Impact | Species | Duration | References |
|---|---|---|---|---|---|
| Atorvastatin, Fluvastatin, Lovastatin, Pitavastatin, Pravastatin, Rosuvastatin, Simvastatin | Statin | Embryogenesis | *Danio rerio, Pimephales promelas* | 96–144 h | [22] |
| Simvastatin | Statin | Development, cardiac function, embryo death | *Danio rerio* | 96 h | [23] |
| Atorvastatin, Fluvastatin, Lovastatin, Pravastatin, Rosuvastatin, Simvastatin | Statin | Development, gene regulation, death | *Danio rerio* | 96 h | [24] |
| Simvastatin | Statin | Embryogensis, biochemical markers, molecular markers | *Danio rerio* | 90 days | [25] |
| Atenolol, Propranolol | Beta-blocker | Bioaccumulation | *Pimephales promelas* | 7 days | [26] |
| Atenolol | Beta-blocker | Lipidomic, metabolomic, behavior response | *Danio rerio* | 7 days | [27] |
| Propranolol | Beta-blocker | Condition Factor, liver-somatic index, sex characteristics | *Pimephales promelas* | 165 days | [28] |
| Bisoprolol, Sotalol | Beta-blocker | Locomotor behavior | *Danio rerio* | 25 min | [29] |

Herein, we report the correlations between several fish condition metrics from multiple fish species along with environmental surface water concentrations of statins and beta-blockers. We collected multiple species of fish at locations upstream, downstream, and near two West Virginia wastewater treatment plants (WWTPs). Using the surface water concentration data, we created a Bayesian linear mixed effects model capable of accounting for the spatial, physiological, and individual pharmaceutical differences between sampling locations and fish genera. The objective of this study was to investigate whether relevant surface water concentrations are negatively correlated to fish health and, if so, at what concentration level.

## 2. Materials and Methods

### 2.1. Study Area

Water samples and fish specimens were collected near 2 wastewater treatment plants located in the towns of Elkins and Weston from 2 rivers in West Virginia: the Tygart Valley River, and the West Fork River. The Tygart Valley River, a major tributary of the Monongahela River, begins in Pocahontas County and flows north, northwest for 215 km,

where it converges with the West Fork River in Fairmont, WV to form the Monongahela River. The Tygart Valley River has a drainage area of 3562 km$^2$ and a mean annual discharge of 10.19 m$^3$/s [30]. In comparison, the West Fork River begins in Upshur County in West Virginia and flows northeast for 149 km where it converges with the Tygart Valley River in Fairmont, WV. The West Fork River has a drainage area of 2280 km$^2$ and a mean annual discharge of 4.80 m$^3$/s (Figure 1) [30]. The Elkins WWTP discharges its effluent into the Tygart Valley River. It has an average discharge of 0.090 m$^3$/s and utilizes grit removal, activated sludge, an oxidation ditch, clarifiers, and UV disinfection for its treatment process. Weston's WWTP facility discharges effluent into the West Fork River with an average discharge of 0.048 m$^3$/s. The Weston WWTP uses a combination of grit removal, extended aeration, and clarifiers with its treatment process. Both WWTPs discharge their treated wastewater via discharge pipe(s) that are located along the bank of the river, creating an outfall area.

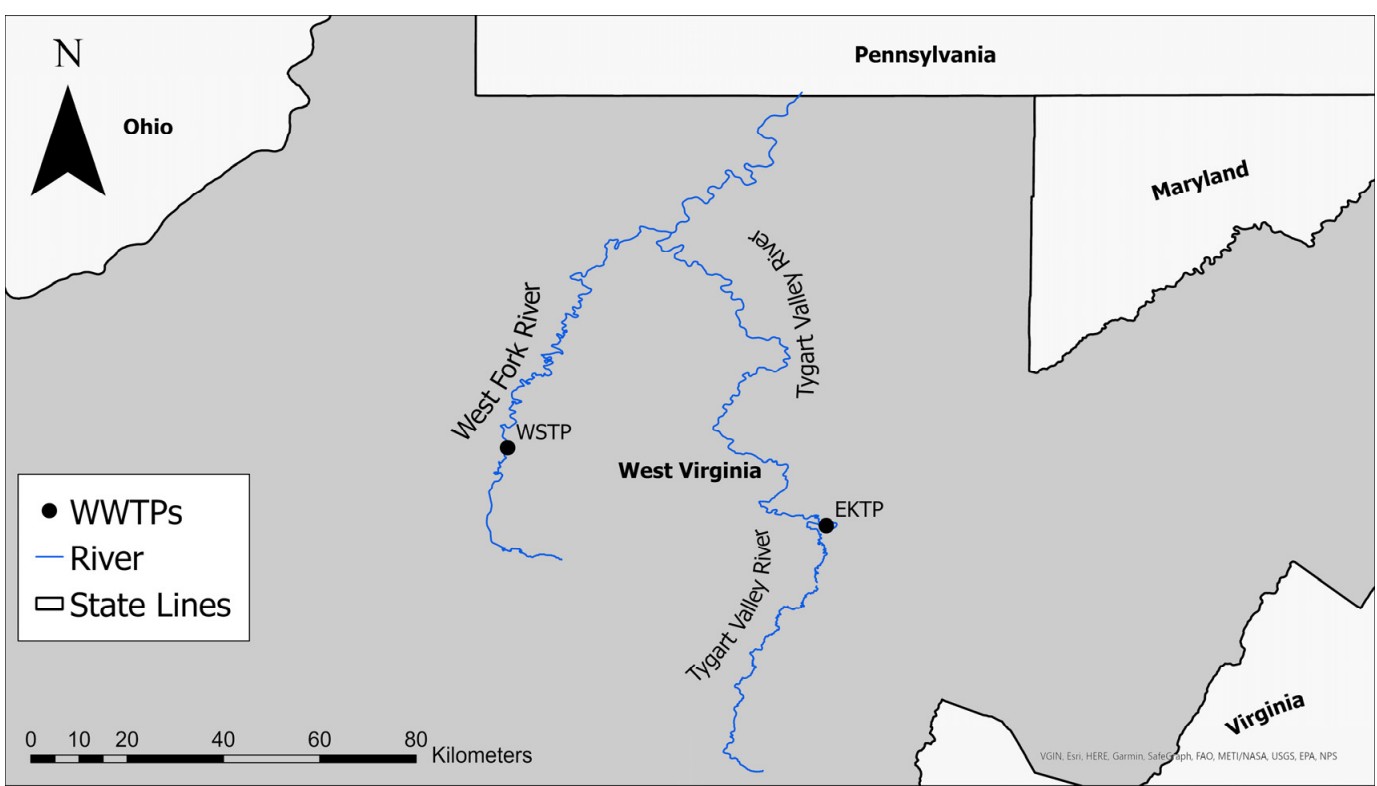

**Figure 1.** Wastewater treatment facility locations on the West Fork and Tygart Valley Rivers in West Virginia.

## 2.2. Surface Water Collection

Effluent samples were collected at each WWTP, and samples were also collected from areas 150 m upstream and downstream from the effluent discharge point of each WWTP (Figure 2). Water samples were collected from each WWTP monthly near the end of the month over a 4-month period (August 2020–November 2020). Each month, a total of 6 samples were collected, 3 samples per WWTP. At the end of the 4-month sampling period, a total of 24 samples had been collected. Water sampling occurred during median flow conditions to simulate the typical flow conditions during the low flow period of the year. Samples were collected during the low flow period to maximize the likelihood of detecting pharmaceuticals of interest.

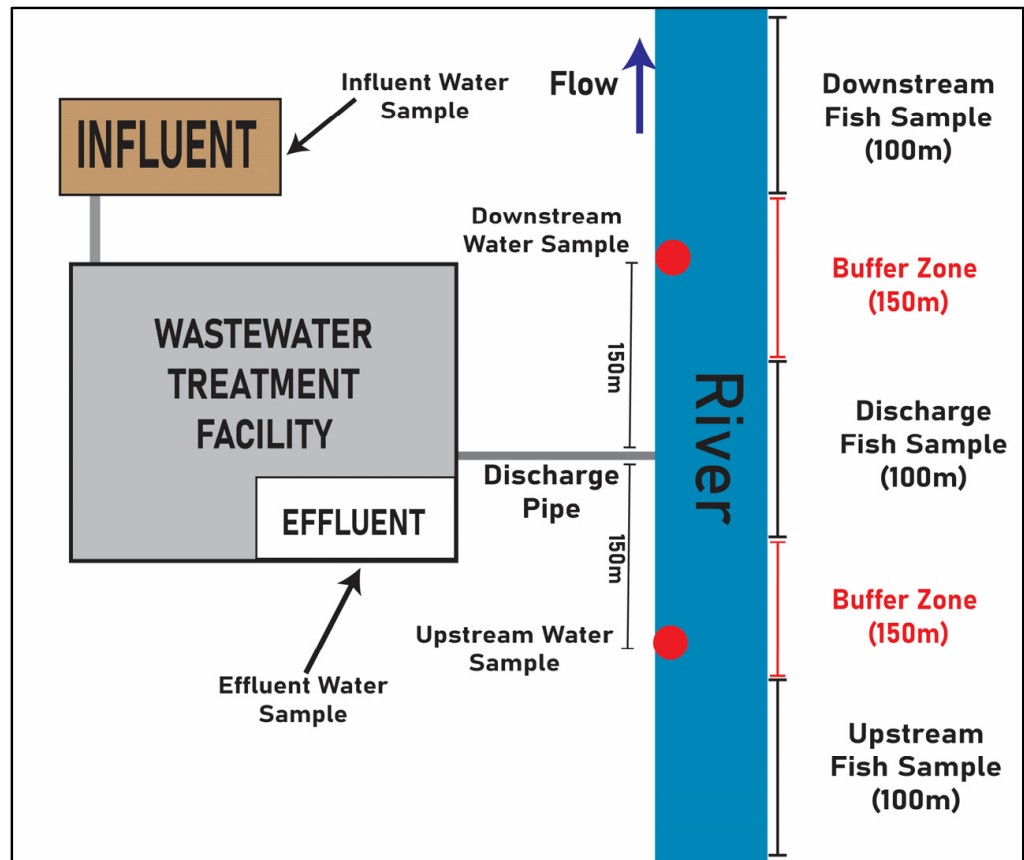

**Figure 2.** Water and fish sampling diagram for collecting water samples and fish specimens at each wastewater treatment plant. The red dots represent water grab samples in the river.

Note that water samples for the effluent sample were collected directly from the effluent outflow inside the WWTP and not directly from the discharge area in the river itself. To account for this, discharge area concentrations were estimated based on a simple dilution Equation (1):

$$C_3 = (C_1 V_1 + C_2 V_2)/V_3 \tag{1}$$

where $C_1$ is the upstream concentration, $V_1$ the volume of water upstream, $V_2$ is the volume of water exiting the effluent discharge pipe, $C_2$ is the concentration of the effluent, $V_3$ is the combined flow of the river upstream of the discharge pipe and the flow of the discharge pipe, and $C_3$ is the estimated concentration in the discharge area.

Sampling sites were visited in a non-random order due to the distance between sites. The effluent samples were collected with HDPE dipper arms, whereas the riverine samples were grab samples collected either by wading or paddling a 10-foot john boat to the middle of the river. Grab samples were collected using 950 mL, pre-cleaned, glass amber bottles and then placed on ice for transport. All collection bottles were rinsed at least twice with sample water prior to final collection, and all samplers were required to wear nitrile gloves, protective eyewear, and surgical masks while collecting and handling, to minimize contamination and exposure to harmful biological agents. Water samples were placed in an ice chest and transported back to the laboratory and refrigerated until ready for analysis.

*2.3. Water Sample Preparation and Analyte Detection*

Water samples were refrigerated at <4 °C for no more than 24 h before undergoing a series of vacuum filtrations using a reusable Nalgene filtration device (Thermo Scientific, Pittsburgh, PA, USA) and a Dry Vacuum Pump (Welch, Mt. Prospect, IL, USA). The samples were filtered through a 47 mm diameter, 13 μm pore non-sterile, hydrophilic cellulose filter.

The second filtration used a 0.45 µm non-sterile, hydrophilic, rinsed nylon or cellulose filter. Filtration was utilized to reduce the likelihood of matrix interference during the analyte detection process.

The following analyte extraction procedure is based off several papers, primarily Gros et al. [31] and U.S. EPA Method 542 [32]. Analyte extracts from water samples, containing internal standards, were compared to external calibration curves of the pharmaceutical standards. Samples and standards are to be analyzed by liquid chromatography with tandem mass spectrometry (LC-MS/MS). This method involves a 5 µL injection onto a C18 UHPLC column (Agilent, Eclipse XDB0C18, 2.1 × 100 mm, with a guard column, Santa Clara, CA, USA), and is held at 35 °C, and gradient separation takes place on a Thermo Scientific Accela UHPLC. Mobile phases were (A) 100% water, 0.1% formic acid and (B) 100% acetonitrile, 0.1% formic acid, and the gradient elution method was: 0 min, 10% B; 7 min, 90% B; 7.1 min, 10% B; 10 min, 10% B. Detection was performed by a Q Exactive Orbitrap mass spectrometer (Thermo Scientific) operating within a parallel reaction monitoring mode to target the protonated form of each pharmaceutical of interest. Overall, this method produced quantitation limits ranging from 0.3 to 0.6 ng/L.

*2.4. Fish Sample Collection and Perseveration*

Fish specimens belonging to 21 different species (*Ambloplites rupestris*, *Ameiurus natalis*, *Cyprinella spoloptera*, *Etheostoma blennioides*, *Etheostoma flabellare*, *Etheostoma nigrum*, *Hypentelium nigricans*, *Labidesthes sicculus*, *Lepomis cyanellus*, *Lepomis gibbosus*, *Lepomis macrochirus*, *Lepomis megalotis*, *Micropterus dolomieu*, *Micropterus salmoides*, *Moxostoma anisurum*, *Moxostoma erythurum*, *Notemigonus crysoleucas*, *Notropis stramineus*, *Pimpephales notatus*, *Pomoxis annularis*, *Pomoxis nigromaculatus*) were collected across 6 sampling locations via raft-based electroshocking, backpack electroshocking, or seines. Fish were collected 200 m upstream and 200 m downstream from the effluent discharge pipe, in addition to fish being collected within 100 m of the effluent discharge pipe (Figure 2). Note that not all species were present at each sampling location. In total, 1018 specimens were collected by the West Virginia Division of Natural Resources (WVDNR) and provided for this study. The species collected are not considered threatened or endangered, while also possessing a stable population within the river systems from which they were collected [33]. Upon collection, fish specimens were euthanized by submergence in a tricaine methanesulfonate (MS-222)-laced live well. Whole specimens were transported back to the laboratory and stored at −5 °C or colder for processing. Individual fish were sexed if mature and measured for total length (mm), weight (g), liver weight (g), and gonad weight (g). Livers and gonads were only extracted from individuals ≥100 mm due to limitations on tissue extraction on smaller specimens.

*2.5. Relative Weight*

We used relative weight (Wr) to assess overall fish condition across many species. Relative weight describes the condition of a fish species within a given population [34]. This method was chosen over relative condition factor because the fish being compared are from the two distinct watersheds. The standard weight, Ws, is calculated to predict the 75th percentile mean weight for a given length with the following length weight relationship Equation (2):

$$Ws = aL^b \tag{2}$$

Standard weight (Ws) equation estimate parameters (a and b) were based on a hierarchical Bayesian length–weight relationship [35] (Table 2). Following a $\log_{10}$ transformation, the equation is readily applied in a linear regression with *a* being the intercept of the regression and *b* being the slope of the regression. After calculating Ws for the given individual, Equation (3) was used to calculate the relative weight (Wr):

$$Wr = 100 \times W/W_s \tag{3}$$

**Table 2.** Fish species and their associated Bayesian length–weight relationship parameters.

| Genus | Species | *a* | *b* |
|---|---|---|---|
| *Lepomis* | *macrochirus* | 0.0141 | 3.13 |
| *Lepomis* | *gibbosus* | 0.0115 | 3.12 |
| *Notemigonus* | *crysoleucas* | 0.0105 | 3.06 |
| *Pimephales* | *notatus* | 0.0069 | 3.21 |
| *Lepomis* | *cyanellus* | 0.0159 | 3.12 |
| *Micropterus* | *salmoides* | 0.0105 | 3.08 |
| *Micropterus* | *dolomieu* | 0.0110 | 3.05 |
| *Notropis* | *stramineus* | 0.0078 | 3.03 |
| *Cyprinella* | *spoloptera* | 0.0083 | 3.07 |
| *Lepomis* | *megalotis* | 0.0141 | 3.12 |
| *Etheostoma* | *blennioides* | 0.0054 | 3.14 |
| *Etheostoma* | *nigrum* | 0.0023 | 3.14 |
| *Etheostoma* | *flabellare* | 0.0054 | 3.14 |
| *Hypentelium* | *nigricans* | 0.0060 | 3.09 |

A Wr value $\geq$ 100 indicated fish in excellent condition, while values of Wr < 100 were increasingly poorer condition (slimmer and less healthy than the average). Application of this condition metric was used to assess potential sub-lethal effects across species and across several sampling sites.

### 2.6. Hepatosomatic Index

The hepatosomatic index (HSI) is a condition index designed to determine the ability of a fish to store lipids in the liver. Previous research has shown that most teleost fish store significant lipids in their livers, while some species also store lipids in their muscles and mesenteric tissues [36]. Ideally, the higher the HSI, the healthier the fish, whereas a lower HSI indicates potential issues with the fish's ability to store crucial energy [37,38]. The equation to calculate HSI is listed below; $W_{liver}$ is the wet weight (g) of the liver and W is the total body weight (g) of the fish. Pharmaceutical impacts to the liver could potentially show up within the HSI.

$$\text{HSI} = 100 \times (W_{liver}/W) \tag{4}$$

### 2.7. Gonadosomatic Index

The gonadosomatic index (GSI) is a measure of the testis and ovaries of fish to determine not only sexual maturity but also to determine indirect energy expenditure towards reproduction [39]. The GSI is calculated by gonad weight (g) divided by total body weight (g) multiplied by 100, as shown in Equation (5) [40]. Comparing the GSI across all sections will show potential gonad development impacts, if there are any.

$$\text{GSI} = 100 \times (W_{gonads}/W) \tag{5}$$

### 2.8. Statistical Analyses

#### 2.8.1. Summary Statistics

Summary statistics, including 95% confidence intervals, were calculated by species for total length (mm), weight (g), Wr, GSI, and HSI at each sampling site. Surface water concentration averages and confidence intervals were generated based on section. These summaries were calculated based on site or river section relative to a WWTP to better highlight variability within the dataset.

#### 2.8.2. ANOVA

We used a series of two-way and three-way ANOVAs in conjunction with post hoc Tukey HSD comparisons to determine if there were significant differences in Wr, HSI, and GSI based on location and genus (Wr) or species (HSI and GSI) [41]. Fish were grouped by genus instead of species for analysis balance, and because not all species were present in

every sample section regarding Wr. HSI and GSI ANOVAs tested for differences between *Lepomis* spp. Additionally, interaction terms, for location and genus, were included with the Wr ANOVAs, while interaction terms of section:species and section:sex were included with the HSI and GSI ANOVAs. Three outliers were removed from 78 observations following diagnostic testing via Cook's Leverage for both the HSI and GSI datasets [42]. The 'stats' [43] package was used to generate the ANOVAs and the post hoc Tukey HSD values. The 'AICmodavg' [44] package was used to compare AIC scores. All analyses were performed in the statistical program R version 4.2.2 [43].

### 2.8.3. Bayesian Linear Mixed Effect Model

Following a two-way ANOVA, we also generated a Bayesian linear mixed effect model to quantify the effects of statins (atorvastatin, simvastatin) and beta-blockers (metoprolol, carvedilol) in surface waters on Wr across section and genus. Similar models for HSI and GSI were not viable due to data limitations (<100 data points). We chose a Bayesian modeling framework due to its power and flexibility after encountering convergence and singularity issues when attempting to use a frequentists approach [45]. The fixed effects included pharmaceutical concentrations, while the random effects included random slopes (pharmaceutical concentrations) and random intercepts (section, genus). Section and genus were structured so that genus was nested within section. Structuring our random effects in this manner enabled us to account for the instances when a genus was absent from a sampling section. The inclusion of site and genus in our random effects enabled the capture variation due to habitat differences within the various sections, and we were able to capture potential physiological differences in the various genera [46,47]. Optimal model selection was based on bayes factor scores [48], marginal (variance explained by fixed effects), and conditional (variance explained by fixed and random effects) coefficients of determination (R2) [45,47], as well as domain knowledge. Models that failed to converge, were unstable, or contained singular boundary issues were discarded. The 'brms' [49] package was used to create and generate model statistics, with tables and graphics being created using the 'sjPlot' [50] and 'ggpubr' [51] packages. All analyses and figures were created in the program R version 4.2.2 [43].

## 3. Results

### 3.1. Pharmaceutical Concentrations

Surface water concentrations for atorvastatin, simvastatin, metoprolol, and carvedilol varied by sampling location and by river/WWTP (Table 3), and all four pharmaceuticals were detected at all sites. Metoprolol consistently had the highest concentration across all samples, with a range of (2.80–9.25 ng/L). Atorvastatin, simvastatin, and carvedilol averages were similar, and ranged between (0.38–2.11 ng/L), (0.19–2.29 ng/L), and (0.40–0.87 ng/L), respectively.

**Table 3.** Average pharmaceutical surface water concentrations with 95% confidence intervals from the two WWTPs on the Tygart Valley and West Fork Rivers.

| River | Section | n | Atorvastatin (ng/L) | Simvastatin (ng/L) | Metoprolol (ng/L) | Carvedilol (ng/L) |
|---|---|---|---|---|---|---|
| Tygart Valley | Upstream | 4 | 0.473 ± 0.151 | 0.268 ± 0.454 | 2.800 ± 1.130 | 0.46 ± 0.102 |
| | Discharge | 4 | 1.827 ± 1.686 | 0.271 ± 0.452 | 21.01 ± 16.79 | 0.487 ± 0.093 |
| | Downstream | 4 | 0.998 ± 0.762 | 0.645 ± 0.650 | 9.250 ± 10.80 | 0.758 ± 0.848 |
| West Fork | Upstream | 4 | 1.313 ± 0.858 | 0.535 ± 0.524 | 4.730 ± 2.849 | 0.87 ± 0.798 |
| | Discharge | 4 | 4.363 ± 2.545 | 0.947 ± 0.739 | 7.049 ± 1.651 | 0.905 ± 0.767 |
| | Downstream | 4 | 2.105 ± 1.102 | 0.535 ± 0.524 | 7.178 ± 5.363 | 0.425 ± 0.313 |

### 3.2. Fish Sample and Condition Metrics

A total of 1018 fish specimens were collected from the Tygart Valley River and the West Fork River, consisting of 14 species across seven genera. The number of specimens caught within each section varied between 118 and 255, with the fewest number of specimens being collected from each of the discharge sampling sections (Table S1). Average relative weight varied between genera; *Pimephales* displayed the highest relative weight ($95 \pm 1.592$), while *Notemignous* displayed the lowest relative weight ($71 \pm 2.703$), with all other genera falling between these values (Table 4). Average HSI values for *L. cyanellus*, *L. gibbosus*, and *L. macrochirus* were 1.17, 1.33, and 1.12, respectively, and average GSI values for those same species were 0.79, 0.64, and 0.67, respectively (Table 5).

**Table 4.** Mean length, weight, and Wr with 95% confidence intervals (CI) for fish genera.

| Relative Weight (Wr) | | | | | | | |
|---|---|---|---|---|---|---|---|
| **Genus** | **N** | **Total Length** | **CI** | **Weight** | **CI** | **Wr** | **CI** |
| | | **(cm)** | | **(g)** | | | |
| *Cyprinella* | 65 | 68.25 | 3.69 | 2.82 | 0.38 | 71.24 | 1.93 |
| *Etheostoma* | 15 | 49.53 | 7.31 | 1.42 | 0.76 | 75.32 | 6.23 |
| *Lepomis* | 332 | 58.95 | 3.40 | 7.88 | 1.68 | 76.97 | 1.21 |
| *Micropterus* | 40 | 71.33 | 10.12 | 7.88 | 5.11 | 88.89 | 4.55 |
| *Notemignous* | 159 | 59.51 | 3.65 | 2.62 | 0.88 | 70.96 | 2.70 |
| *Notropis* | 94 | 38.89 | 3.18 | 0.67 | 0.20 | 80.45 | 2.90 |
| *Pimephales* | 313 | 43.32 | 1.17 | 0.76 | 0.09 | 95.01 | 1.59 |

**Table 5.** HSI and GSI indices for 3 species with 95% confidence intervals (CI). Differences in sample size between Wr and the HSI and GSI is due to specimen size limitations regarding liver and gonad tissue extraction.

| Hepatosomatic Index (HSI) | | | | | | | |
|---|---|---|---|---|---|---|---|
| **Species** | **Sex** | **Total Weight** | **CI** | **Liver Weight** | **CI** | **HSI** | **CI** |
| | **(F/M)** | **(g)** | | **(g)** | | | |
| *L. Cyanellus* | 13/11 | 17.14 | 4.50 | 0.19 | 0.05 | 1.17 | 0.15 |
| *L. Gibbosus* | 17/15 | 30.65 | 8.35 | 0.38 | 0.09 | 1.33 | 0.10 |
| *L. Macrochirus* | 10/9 | 29.00 | 10.16 | 0.29 | 0.07 | 1.12 | 0.11 |
| **Gonadosomatic Index (GSI)** | | | | | | | |
| **Species** | **Sex** | **Total Weight** | **CI** | **Gonad Weight** | **CI** | **GSI** | **CI** |
| | **(F/M)** | **(g)** | | **(g)** | | | |
| *L. Cyanellus* | 13/11 | 17.14 | 4.50 | 0.12 | 0.04 | 0.79 | 0.21 |
| *L. Gibbosus* | 17/15 | 30.65 | 8.35 | 0.15 | 0.03 | 0.64 | 0.12 |
| *L. Macrochirus* | 10/9 | 29.00 | 10.16 | 0.17 | 0.07 | 0.67 | 0.24 |

### 3.3. ANOVA Results

#### 3.3.1. Hepatosomatic Index

A three-way ANOVA compared the effect of sampling section, species, and sex with interaction terms including 'section:species' and 'section:sex'. The HSI ANOVA indicated nearly significant differences in mean HSI between species ($F_{(2, 64)} = [3.029]$, $p = 0.055$), as well as potential differences based on sex ($F_{(1, 64)} = [3.808]$, $p = 0.055$) (Table 6, Figure 3). Tukey's HSD test of multiple comparisons for the HSI ANOVA showed the largest difference among species occurred between *L. macrochirus* and *L. gibbosus* ($-0.209$, $[-0.425, 0.008]$, $p = 0.061$). Meanwhile, the difference between male and female ($-0.138$, $[-0.282, 0.006]$) was also nearly significant ($p = 0.061$).

**Table 6.** ANOVA results for HSI differences based on sampling section, sex, species, and their associated interactions.

| | Df | Sum Squared | Mean Squared | F-Value | *p*-Value |
|---|---|---|---|---|---|
| **Response: HSI** | | | | | |
| Section | 2 | 0.137 | 0.069 | 0.709 | 0.496 |
| Sex | 1 | 0.369 | 0.369 | 3.808 | 0.055 |
| Species | 2 | 0.587 | 0.293 | 3.029 | 0.055 |
| Section:Sex | 2 | 0.030 | 0.015 | 0.156 | 0.856 |
| Section:Species | 3 | 0.430 | 0.143 | 1.480 | 0.228 |
| Residuals | 64 | 6.201 | 0.097 | | |

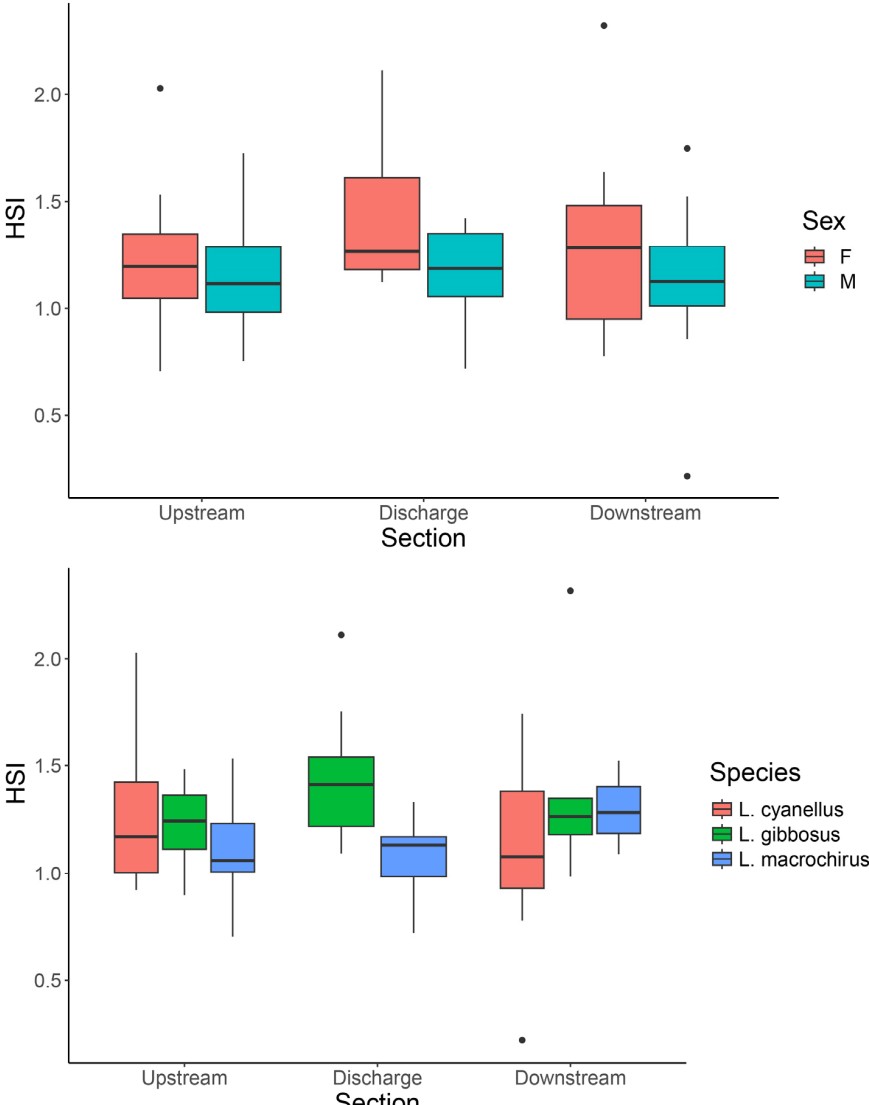

**Figure 3.** HSI ANOVA results for differences in sex and species with sampling section included.

### 3.3.2. Gonadosomatic Index

Differences in mean GSI was compared by a three-way ANOVA based on the effects of sampling section, species, and sex without an interaction term. The GSI ANOVA indicated only significant differences in mean GSI based on sex (F(1, 64) = [63.118], *p* < 0.001) (Table 7, Figure 4). Tukey's HSD test showed a large difference in sex (−0.6038, [−0.759, −0.449], *p* < 0.001).

**Table 7.** ANOVA results GSI differences based on sampling section, sex, species, and their associated interactions.

| | Df | **Response: GSI** Sum Squared | Mean Squared | F-Value | *p*-Value | |
|---|---|---|---|---|---|---|
| Section | 2 | 0.165 | 0.083 | 0.733 | 0.484 | |
| Sex | 1 | 7.103 | 7.103 | 63.118 | <0.001 | *** |
| Species | 2 | 0.314 | 0.157 | 1.396 | 0.255 | |
| Section:Sex | 2 | 0.267 | 0.134 | 1.188 | 0.312 | |
| Section:Species | 3 | 0.376 | 0.125 | 1.114 | 0.350 | |
| Residuals | 64 | 7.203 | 0.113 | | | |

Note: *** Indicates high level of significance.

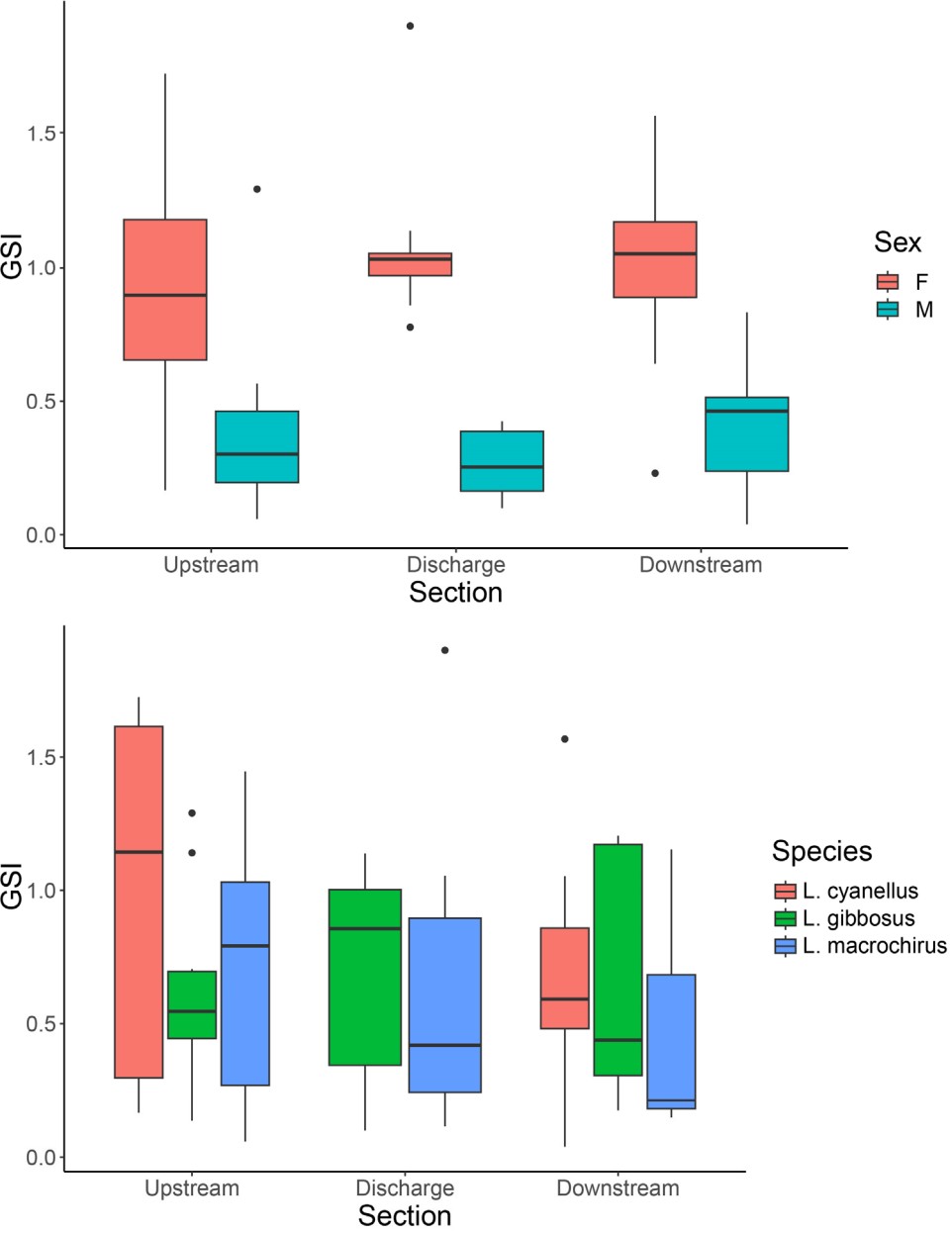

**Figure 4.** GSI ANOVA results for differences in sex and species with sampling section included.

### 3.3.3. Relative Weight

A two-way ANOVA compared the effect of sampling section and genus, including an interaction term, on relative weight. The two-way ANOVA revealed a significant difference in mean relative weight based on genus (F(6, 998) = [88.794], $p < 0.001$) and the interaction term of genus and sampling section (F(11, 998) = [6.584], $p < 0.001$) (Table 8, Figure 5). Tukey's HSD test for multiple comparisons revealed a plethora of significant ($p < 0.05$) differences between various genera and the interaction between genera and section (Tables S2 and S3).

**Table 8.** ANOVA results relative weight differences based on sampling section, sex, species, and their associated interactions.

| | **Response: Relative Weight** | | | | | |
|---|---|---|---|---|---|---|
| | **Df** | **Sum Squared** | **Mean Squared** | **F Value** | **$p$ Value** | |
| Section | 2 | 54 | 26.9 | 0.155 | 0.8566 | |
| Genus | 6 | 92.60 | 15,433 | 88.794 | <0.0001 | *** |
| Section:Genus | 11 | 12.59 | 1114 | 6.584 | <0.0001 | *** |
| Residuals | 998 | 17,346 | 173.8 | | | |

Note: *** Indicates high level of significance.

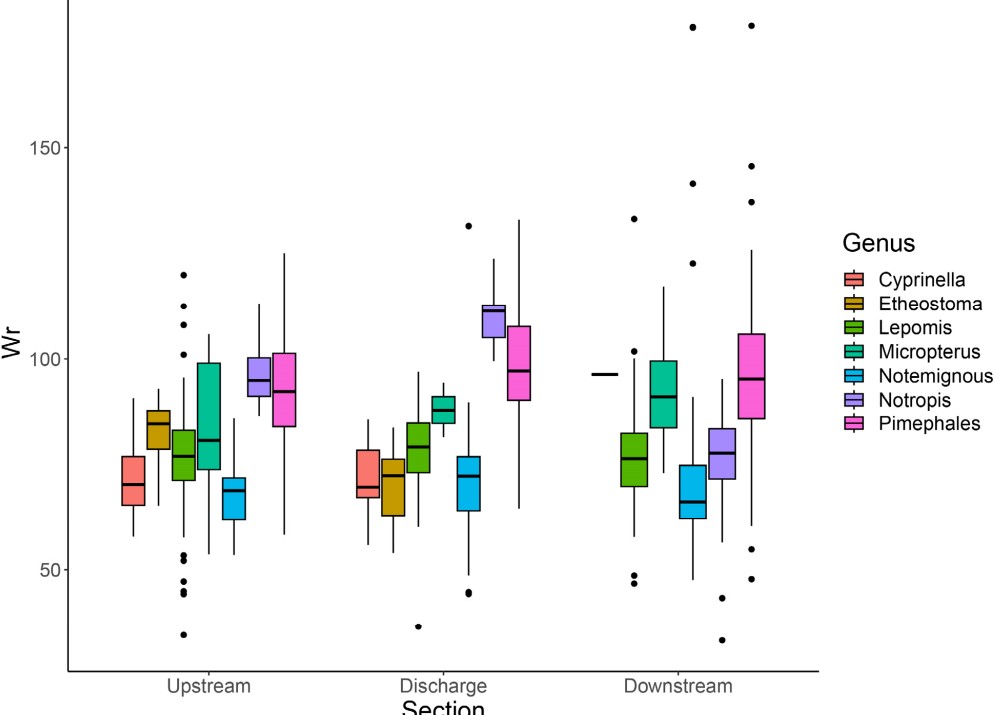

**Figure 5.** Relative weight ANOVA results for differences in genus with sampling section included.

### 3.4. Bayesian Linear Mixed Effects Model

Following model selection, the optimal model for estimating relative weight changes related to pharmaceutical surface water concentrations included random intercepts of Genus nested within sections and random slopes of pharmaceutical concentration (Figure 6). None of the pharmaceuticals displayed a significant effect on relative weight, with Atorvastatin and Metoprolol showing small positive effects on relative weight (0.207 and 0.052), while Simvastatin and Carvedilol displayed small negative effects (−0.139 and −0.262) (Table 9). Together, the fixed and random effects explained 37.6% of the total variance (conditional $R^2$ = 0.376) and the pharmaceutical surface water concentrations; the fixed effects by themselves accounted for 0.7% of variance (marginal $R^2$ = 0.007).

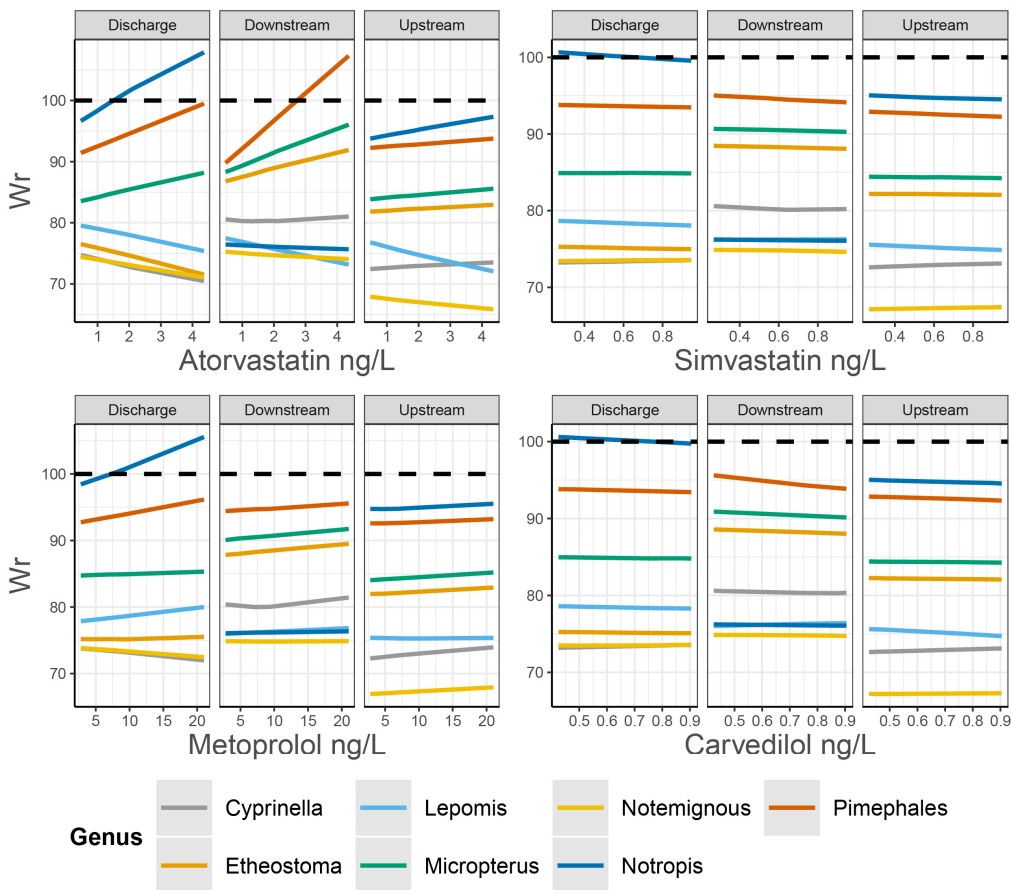

**Figure 6.** Bayesian linear mixed effects graphic showing response of individual genera to various concentrations of pharmaceuticals across all sampling sections. Black dashed line is representative of a healthy relative weight value of 100.

**Table 9.** Bayesian linear mixed effects model results for relative weight as it relates to surface water concentrations of atorvastatin, simvastatin, metoprolol, and carvedilol.

| Predictors (Fixed Effects) | Relative Weight | |
| --- | --- | --- |
| | **Estimates** | **CI (95%)** |
| Intercept | 81.66 | 74.11–89.23 |
| Atorvastatin | 0.207 | −1.371–1.845 |
| Simvastatin | −0.139 | −2.072–1.784 |
| Metoprolol | 0.052 | −0.533–0.584 |
| Carvedilol | −0.262 | −2.164–1.682 |
| Random Effects | | |
| $\sigma^2$ | 149.78 | |
| $\tau_{00}$ Section:Genus | 126.30 | |
| $\tau_{11}$ Section:Genus.Ator | 13.41 | |
| $\tau_{11}$ Section:Genus.Simv | 45.34 | |
| $\tau_{11}$ Section:Genus.Meto | 0.26 | |
| $\tau_{11}$ Section:Genus.Carv | 42.23 | |
| ICC | 0.52 | |
| $N_{Section}$ | 3 | |
| $N_{Genus}$ | 7 | |
| Observations | 1018 | |
| Marginal $R^2$/Conditional $R^2$ | 0.007/0.376 | |

## 4. Discussion

We observed differences in liver, gonad, and relative weight based on species/genus and sex in riverine waters around two WWTP within the Tygart Valley River and West Fork River, but none of these were consistently related to the selected pharmaceutical levels. *Lepomis* spp. displayed nearly significant differences in liver condition (assessed via HSI) based on individual species and sex, with no differences observed based on sampling section. In addition, *Lepomis* spp. displayed differences in gonad condition (via GSI) based on sex, but not species or sampling section. Significant differences in relative weight were identified based on seven genera and the interaction of genera within a given section, but section by itself was not significant. Further analysis of relative weight revealed that section and genus were significant random effects; however, all four pharmaceuticals (atorvastatin, simvastatin, metoprolol, and carvedilol) did not appear to have a significant effect on relative weight at any of the observed concentrations. Furthermore, the marginal effects from the mixed linear effects model were a mixed bag of positive (atorvastatin and metoprolol) and negative effects (simvastatin and carvedilol).

Differences in HSI between species and sex near WWTP's could be the result of several factors, including phylogenetic differences, pollution tolerance, and indirect dietary factors. Phylogenetic differences among the species we observed could lead to significant variation in the HSI scores even if these species belong to the same genus [52]. Similarly, different species of *Lepomis* have varying degrees of pollution tolerance, which could explain some of the observed differences in HSI [53]. Several studies have demonstrated how WWTP plant effluent negatively impacts certain species to a greater extent, compared to more tolerant species [54–56]. In addition to chemical contaminants, thermal pollution can also impact liver physiology by disrupting the physiological processes associated with temperature acclimation [57], and during November, the water exiting the WWTP is likely warmer than the native river temperature. The differences in effluent temperatures and native temperatures are likely to implore stress on all fish, with some species acclimating more efficiently than others [58,59]. The observed differences in HSI scores based on sex of the individual fish is likely linked to vitellogenesis in the liver which causes the liver of female fish to increase in size [60]. Vitellogenesis occurs leading up to a fish's spawning window and, in some cases, will occur leading up to winter, but is seasonal in nature and is dependent upon a species spawning period. There is the possibility that endocrine-disrupting compounds from the WWTP were stimulating vitellogenesis in the female fish, leading to the increase in liver size compared to the males [61]. Further testing would be required to prove this hypothesis for the samples used in this study, but it should also be considered for future studies. Based on the receiving waters and the treatment processes of both WWTP, it is reasonable to assume that endocrine disrupting compounds have not been removed and are entering the river systems. However, no differences in HSI were observed based on sample section, which could indicate that the difference in pharmaceutical concentrations provided by the nearby WWTP did not have a noticeable effect on the livers of the fish. It is worth noting that there is the distinct possibility that since fish are mobile organisms, they do not spend their whole lives in any single section for 4 months, let alone an entire year. The mobility of the fish could explain why we did not observe differences in HSI based on section.

Like the HSI scores, differences in the GSI scores were detected based on the sex of the fish and are likely tied to inherent differences in mature male and female GSI values and influential factors stemming from nearby WWTP effluent. *Lepomis* spp. are typically summer spawners and, since the specimens were collected in November, both male and female specimens were likely focused on somatic growth and not gonadic growth [62]. However, as adults, *Lepomis* gonads possess inherent size proportion differences even during periods of primarily somatic focused growth [63]. To account for this, we included a sex and section interaction term, but it was not significant. Previous research has shown, however, that differences in GSI are easier to detect right before or during spawning season, which could account for the lack of a significant difference between sampling sections [64].

Differences in relative weight were observed between genera and the interaction between genera and sample section, indicating that certain groups of fish are experiencing varying degrees of impacts to their growth based on their genus and their location. Previous research has established that pollution tolerance varies by genus and species, and the variation in pollution levels upstream and downstream from the WWTP likely explain some of these differences [65–67]. More specifically, fish near the discharge and downstream are exposed to a variety of pollutants, which can cause endocrine disrupting effects, genotoxicity effects, immunotoxicity effects, and behavioral changes, to name a few [54,55,66,68]. There is also the consideration of indirect effects on fish health through changes in food availability/quality. Macroinvertebrates and microscopic organisms inhabiting the same area experience the same pollution levels, but with a greater sensitivity which, in turn, may affect their growth and abundance, therefore leading to changes in available food and food quality for fish. [8,21,69]. Furthermore, the marginal effects produced from the linear mixed effects model for atorvastatin, simvastatin, metoprolol, and carvedilol were not statistically significant, but the overall relative weight based on the mixed effect model showed that fish within both study sites were likely stressed upstream and downstream of the WWTPs. We observed pharmaceutical concentrations for all four pharmaceuticals upstream of the WWTPs, indicating the widespread prevalence of these pharmaceuticals. In turn, that meant that non-exposed fish were never observed or compared to fish that were exposed to these compounds. Based on the known pharmacology of the cardiovascular medicines of interest and the marginal effect sizes, it is likely that there was some type of effect, but it is difficult to tell if it was beneficial or harmful based on the relative weight metrics.

While the information provided in this study is valuable, it is not without its limitations, and areas for either improvement or expansion are available. One of the biggest limitations encountered in this study was that the experimental design is purely observational, which can lead to bias and difficulty sourcing unknown variance in the dataset [70]. In the same vein, it was difficult to account for either fish immigration or emigration due to physical constraints within the rivers themselves, as well as any type of survivorship biases within our samples [71,72]. Other contaminants present within the WWTP effluent could also be responsible for some of the negative impacts we observed [54,65,73]. Finally, the last issue was that the metrics chosen to assess fish health are a small set of metrics used to assess fish, and that other metrics might be more effective for future studies. A list of exhaustive alternatives can be found in Murphy et al. [74].

Future work on cardiovascular medicine wastes will need to incorporate laboratory studies and field studies to capture seasonal changes, as well as community-wide impact assessments. Controlled lab experiments will allow for testing complex mixtures of pharmaceuticals at different environmental concentrations while minimizing outside factors. Uncertainty analyses should also be included with lab experiments to better highlight the contribution of error in future experiments. Furthermore, seasonal variation, as it relates to spawning times for different species, can be studied more efficiently in a laboratory setting to determine if these cardiovascular pharmaceuticals impact reproductive systems. Applying the principles of ecotoxicology, the use of community-wide assessments that include fish, macroinvertebrates, and lower-level lifeforms can also help elucidate if exposure to environmentally relative concentrations of cardiovascular medicines leads to significant changes in aquatic communities [75]. Alternative methods of identifying sublethal responses in fish species should include genetic response testing, as well as testing for mechanistic effects of chemicals at the cellular and molecular level, which can then be extrapolated to the wider community [75]. Detailed histology and tissue analyses combined with these tests will assist managers in determining the scale of impact from cardiovascular medicinal waste. Ideally, the results and measurements from these future works in conjunction with more accurate loading assessments could be applied to comprehensive life cycle assessments (LCA) for the various statins and beta-blockers currently in circulation [76]. However, the uncertainty observed within our study will need to be reduced prior to inclusion in an LCA due to the sensitivity of LCAs to uncertainty [77].

Within the limitations of this study at the current environmental concentrations, there is a limited effect from atorvastatin, simvastatin, metoprolol, and carvedilol on fish liver, gonads, and overall condition. Streams and rivers not dominated by WWTP discharge still contain detectable and quantifiable concentrations of these pharmaceuticals, indicating widespread distribution and potentially ubiquitous impacts at various levels. However, streams that are predominantly made up of WWTP effluent discharge will contain greater concentrations, and thus be at greater risk for sublethal impacts. The methods from this study can be applied to any river or stream system to assess fish assemblage conditions relative to pharmaceutical inputs, but care must be taken to assess the pharmaceutical mixture in the surface waters before drawing any conclusions. Managers of effluent-dominated systems can use this methodology in conjunction with other methods to better define and understand the extent of statins, beta-blockers, or other pharmaceutical impacts on their fisheries. Further refinements in this methodology will include the addition of tissue concentration testing and comparing tissue concentrations to various metrics, such as those in this study to further understand potential impacts to fish growth, condition, and survival. Additional considerations should also include surrounding land use that may cause alterations to flow regimes or other anthropogenic impacts [78]. Pharmaceutical wastes by themselves may be unlikely to cause fish kills, but pharmaceutical wastes impacts compounded with other anthropogenic impacts can potentially cause substantial damage to fisheries and aquatic communities.

## 5. Conclusions and Prospects

Pharmaceuticals and personal care products used in the treatment of cardiovascular disease are constantly being discharged into surface waters across the globe. These cardiovascular medicines have the potential to impact aquatic organisms due to their ability to directly target biological processes. We assessed several health metrics for fish that inhabit areas within the vicinity of two wastewater treatment facilities in West Virginia, and compared the relative weight of these fish to surface water concentrations of atorvastatin, simvastatin, metoprolol, and carvedilol. Differences in hepatosomatic and gonadosomatic indices were detected based on sex and/or species, but not based on location relative to the wastewater treatment facility. Comparatively, differences in relative weight were observed based on species and location with a Bayesian linear mixed effect model showing limited positive and negative effects to relative weight based on surface water concentrations. While there appears to be some sub-lethal response to these pharmaceuticals, it is difficult to ascertain the risk associated with these changes. Given these results, further research is required to effectively elucidate changes within specific tissues, as well as investigating seasonal effects associated with prolonged, year-round exposure to these substances. Additionally, development of standardized in-tissue concentration measurement methods is required to better compare fish health metrics to surface water concentrations and exposure regimes.

**Supplementary Materials:** The following supporting information can be downloaded at: https://www.mdpi.com/article/10.3390/w15203536/s1, Table S1: Fish sampling results separated by river and sample section; Table S2: Tukey HSD post hoc comparisons of differences in Wr based on genus; Table S3: Tukey HSD post hoc comparison of differences in Wr based on the 'section:genus' interaction term.

**Author Contributions:** Conceptualization, J.W.K. and K.J.H.; Data curation, J.W.K.; Formal analysis, J.W.K.; Investigation, J.W.K.; Methodology, J.W.K.; Resources, K.J.H.; Supervision, K.J.H.; Validation, J.W.K. and K.J.H.; Visualization, J.W.K.; Writing—original draft, J.W.K.; Writing—review and editing, J.W.K. and K.J.H. All authors have read and agreed to the published version of the manuscript.

**Funding:** This research was funded by the United States Geological Survey and West Virginia University, and the APC was funded by West Virginia University.

**Institutional Review Board Statement:** This research was conducted using approved protocols and under direct supervision of the West Virginia Department of Natural Resources. All animals were handled in accordance with West Virginia and United States policies on animal handling for research purposes.

**Data Availability Statement:** The data presented in this study are available on request from the corresponding authors. The data are not publicly available due to ongoing research with the dataset.

**Acknowledgments:** The authors acknowledge Callee Walsh for analyzing the water samples, the West Virginia BioNano Research Facility for providing expertise and equipment, The WVDNR for collection of the fish, The Hartman Lab at West Virginia University, and the undergraduate student workers who helped dissect and process fish.

**Conflicts of Interest:** The authors declare no conflict of interest.

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
