# Peer review of "The Potential Impacts of Statins and Beta-Blockers on West Virginia Ichthyofauna"

_water, doi:10.3390/w15203536_

Round 1

Reviewer 1 Report

This paper needs major revision before publication in “Water”. 1)      The abstract is a bit long. 2)      Please add three relevant keywords. 3)      Please add a Table of Abbreviations/Nomenclatures/Symbols. 4)      Please avoid reference lumping. 5)      Some paragraphs are too long. This negatively impacts the structure of the manuscript. 6)      Please avoid having heading after heading with nothing in between; either merge your headings or provide a small paragraph in between. 7)      The novelty/originality of the paper should be more effectively established. It would be advisable to add a Table to the “Introduction” section, tabulating the latest research works in the field to highlight the novelty of the present work accordingly. 8)      An uncertainty analysis should be performed and incorporated into experimental studies.

9)      Avoid using abbreviations in the headings.

10)  When a term is abbreviated, it should appear in full the first time it is used. This has not been fully observed throughout the manuscript.

11)  Please ensure consistency in presenting units, either using the “/” or “-1” style.

12)  The sections “Results” and “Discussion” should be merged.

13)  Limitations of the study should be included and discussed. 14)  Add practical implications of the study. 15)  The obtained results have not been sufficiently compared with the published data. Please add a Table in the “Results and Discussion” section to address this issue. 16)  The obtained results are promising. Nevertheless, future studies should further investigate the 16)results presented using advanced sustainability assessment tools, including life cycle assessment, as elaborated in recent works such as “Environmental life cycle assessment of biodiesel production from waste cooking oil: A systematic review”, “The role of sustainability assessment tools in realizing bioenergy and bioproduct systems”, and “Environmental life cycle assessment of different biorefinery platforms valorizing municipal solid waste to bioenergy, microbial protein, lactic and succinic acid”, Authors can briefly discuss this need using works such as the example provided, but not necessarily limited to that, and highlight this future research need. 17)  Please change “Conclusions” to “Conclusions and Prospects”. This part simply presents the results obtained throughout the study. Please extend this section by including more research directions.

Author Response

Please see the attached table where we list the numbered comments and our responses to the comments. Thank you.

Reviewer 2 Report

Pharmaceuticals, pharmaceutical metabolites, and other personal care products are found in surface and ground waters throughout the world including the streams and rivers in the United States. Therefore, ecotoxicological effects of these pharmaceutical pollutants on aquatic organisms are quite important. This study provides basic data to these matters. This study is worthy of publication in WATER.

Please correct the following points

1)     Please provide a section on animal ethics and describe it.

2)     A summary scheme or graphic abstract would emphasize the results of this study.

Author Response

Please see the attached PDF where we list the comments from reviewer 2 and our responses.  Thank you.

Round 2

Reviewer 1 Report

This paper can be published in its present form.